# PPLLaVA: Varied Video Sequence Understanding With Prompt Guidance

**Shangkun Sun**[1,2*]    **Ruyang Liu**[1,2*]    **Haoran Tang**[1,2]    **Yixiao Ge**[3]    **Haibo Lu**[1]
**Jiankun Yang**[1†]    **Chen Li**[4†]
[1]Peng Cheng Laboratory    [2]Peking University    [3]XPeng Inc.
[4]Ministry of Industry and Information Technology of the People's Republic of China,
China Electronics Standardization Institute, Beijing, China
{sunshk@stu, ruyang@stu, hrtang@}pku.edu.cn    lichen@cesi.cn
{luhb, yangjk}@pcl.ac.cn    geyixiao831@gmail.com

## Abstract

In the past year, video-based large language models (Video LLMs) have achieved impressive progress, particularly in their ability to process long videos through extremely extended context lengths. However, this comes at the cost of significantly increased computational overhead due to the massive number of visual tokens, making efficiency a major bottleneck. In this paper, we identify the root of this inefficiency as the high redundancy in video content. To address this, we propose a novel pooling strategy that enables aggressive token compression while retaining instruction-relevant visual semantics. Our model, Prompt-guided Pooling LLaVA (PPLLaVA), introduces three key components: a CLIP-based visual-prompt alignment module that identifies regions of interest based on user instructions, a prompt-guided pooling mechanism that adaptively compresses the visual sequence using convolution-style pooling, and a clip context extension module tailored for processing long and complex prompts in visual dialogues. With up to 18× token reduction, PPLLaVA maintains strong performance across tasks, achieving state-of-the-art results on diverse video understanding benchmarks—ranging from image-to-video tasks such as captioning and QA to long-form video reasoning—while significantly improving inference throughput.

## 1 Introduction

Recent advances in Multimodal Large Language Models (MLLMs), such as the LLaVA series Liu et al. (2024b); Li et al. (2024a), Qwen-VL series Yang et al. (2024); Bai et al. (2025), and InternVL series Chen et al. (2024a); Zhu et al. (2025), have led to unified models capable of handling both images and videos. A prevalent approach to modeling videos is to directly feed all frame-wise visual tokens into the LLM. Thanks to the extended context lengths supported by advanced MLLMs, they can effectively capture temporal dependencies over long video sequences, making them effective for long video understanding. However, this strategy also introduces substantial computational overhead due to the massive number of video tokens, posing a major challenge for real-time or resource-constrained applications.

To address this, various token reduction strategies have been explored. Early methods employed temporal average pooling to compress frame sequences (Li et al., 2023b; Maaz et al., 2023; Luo et al., 2023; Liu et al., 2024d), but at the cost of losing temporal dynamics. More recent efforts for long video modeling introduce specialized structures like visual memory (Ren et al., 2024; Zhang et al., 2024a; Zhou et al., 2024) or adaptive keyframe selection Wang et al. (2024b;c); Shen et al. (2024); Liu et al. (2025), which improve long video handling but often lack flexibility for short videos. In contrast, conditional token pooling or aggregation (Li et al., 2023d; Xu et al., 2024a; Jin et al., 2023) offers a more general solution, enabling substantial token compression while preserving

---

*Equal contribution.
†Corresponding authors.

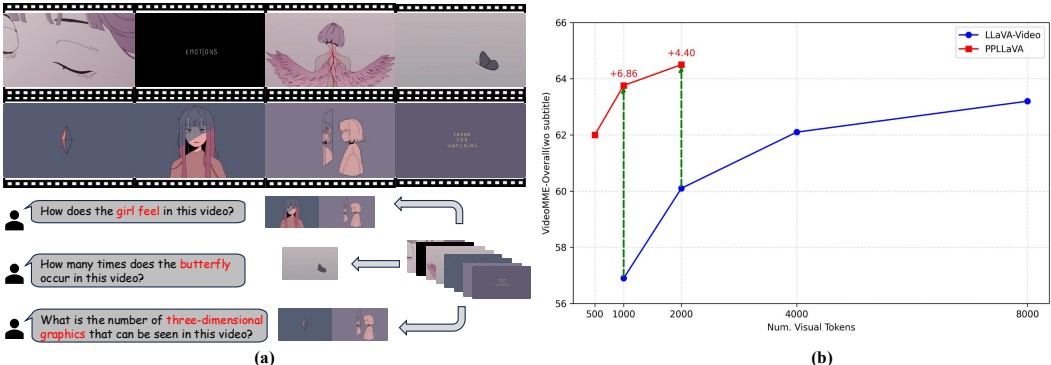

Figure 1: (a) An instance from VideoMME (Fu et al., 2024). The crucial information pertains to only a small portion of the video for different questions. (b) Performance comparison of PPLLaVA-LLaVA-Video with LLaVA-Video baseline on VideoMME-Overall (wo subtitles), which shows the number of visual tokens input to the LLM and the corresponding model performance.

spatiotemporal structure, and is thus increasingly favored in recent baseline MLLM designs Zhang et al. (2024c); Bai et al. (2025).

However, pooling inevitably leads to performance degradation compared to using the full set of visual tokens. As a result, most existing models adopt conservative pooling scales, typically reducing the sequence length by only a factor of 4, to strike a balance between efficiency and performance. But is it possible to further reduce token length without sacrificing modeling capabilities? We believe the key lies in the intrinsic redundancy of videos. As shown in prior work (Han et al., 2022; Liu et al., 2023a; Ma et al., 2022; NVIDIA et al., 2026), key visual information is often concentrated in a few frames—especially in long videos. For video LLMs, this is even more evident: as illustrated in Fig. 1(a), user instructions may only pertain to a small portion of the video, rendering much of the remaining content irrelevant. If we can effectively extract instruction-relevant visual features while compressing the token length, we may maintain or even enhance performance. In this context, the design of vision-language mapping modules becomes critical. Models using Q-Formers Li et al. (2023a); Dai et al. (2023) perform token compression by converting rich visual inputs into a small set of query tokens while enabling interaction with the instruction text. However, modern MLLMs such as LLaVA series Li et al. (2024a), Qwen-VL series Bai et al. (2025), and InternVL Chen et al. (2024a) series have moved away from Q-Formers in favor of simpler architectures like linear projections or MLPs, which are easier to train and more efficient in inference. This raises an important question: Can we develop a pooling strategy that achieves the token efficiency and instruction alignment of Q-Formers, while retaining the simplicity and scalability of current mainstream models?

To this end, we propose Prompt-guided Pooling LLaVA (PPLLaVA), a novel method that combines visual token pooling with instruction-aware visual feature extraction. Specifically, PPLLaVA first identifies prompt-relevant visual representations through fine-grained vision-prompt alignment. It then uses the resulting prompt-vision relevance map as a 3D convolutional kernel to compress visual tokens into any desired three-dimensional size, based on a specified output resolution or stride. In addition, recognizing that CLIP pretraining imposes a limited text context length—and that training video LLMs, especially for multi-turn dialogues, demands extended textual input—PPLLaVA incorporates asymmetric positional embedding extensions to enhance the model's text encoding capacity. As a result, PPLLaVA effectively extracts instruction-relevant visual features from both long-form and short-form textual prompts while significantly reducing the visual token length. It achieves over 90% token compression, supports ultra-long video inputs, and simultaneously enhances performance on short-video tasks.

Experiments on the latest multimodal LLM benchmarks have validated the superiority of our method: PPLLaVA has achieved top results across a wide range of test sets, including NextQA Xiao et al. (2021a), EgoSchema, ActivityNet (Caba Heilbron et al., 2015), VCG Bench (Maaz et al., 2023), MVBench (Li et al., 2023c), LongVideoBench (Wu et al., 2024), and Video-MME (Fu et al., 2024). These benchmarks encompass tasks such as video question answering, detailed video captioning, and video multiple-choice questions, with video lengths ranging from seconds to hours. We

conducted post-training with the integration of PPLLaVA on different VLM base models, including LLaVA-Next, LLaVA-Video, and InternVL3. These VLMs vary in terms of training token scales and visual encoders, yet PPLLaVA consistently yields further improvements, demonstrating the generalization capability of our model. More importantly, PPLLaVA significantly improves efficiency while achieving better performance. As shown in Fig. 1(b), compared to the baseline LLaVA-Video, PPLLaVA achieves superior performance with only one-quarter of the token count. When the number of tokens is aligned, the advantage of PPLLaVA becomes even more pronounced, outperforming by 6.86% and 4.4% at 1000 and 2000 tokens, respectively. It is worth noting that the performance of PPLLaVA does not solely rely on the semantic alignment provided by CLIP. On the contrary, CLIP-based encoders offer strong initialization, while the model learns during subsequent training how to adaptively extract critical visual information. As a result, PPLLaVA still achieves outstanding performance even on VideoChatGPT-Bench, where summarization and caption-style problems are prevalent without informative user questions.

## 2  RELATED WORK

**Multimodal Large Language Models.** Image-domain pretrained models have long served as the foundation for video understanding (Carreira & Zisserman, 2017; Luo et al., 2022; Liu et al., 2023b; Ye et al., 2026; Peng et al., 2026; Sun et al., 2025a;b). This is partly due to the inherent similarities between image and video modalities and partly because image pretraining datasets offer a level of quality, quantity, and diversity that video datasets often lack. Early MLLMs, such as the LLaVA Liu et al. (2024a;b) and BLIP series Li et al. (2023a); Dai et al. (2023), were typically trained with images during SFT, thus requiring additional investigation into how to transfer them to the video domain. In contrast, the latest open-source MLLMs—such as LLaVA-OneVision Li et al. (2024a), Qwen2.5-VL Bai et al. (2025), and InternVL3 Zhu et al. (2025), have already integrated image, multi-image, and video training. Thanks to their support for extremely long contexts (often exceeding 16k tokens), inputting long videos is no longer a challenge. However, the overhead introduced by accommodating long videos through extremely long contexts not only demands substantial computational resources but also hinders lightweight video understanding and deployment on resource-constrained devices.

**Video LLMs.** In the past year, Video LLMs have experienced rapid development. Early Video LLMs (Li et al., 2023b; Zhang et al., 2023; Luo et al., 2023; Maaz et al., 2023; Liu et al., 2024d) typically used average pooling to process video sequences with Image LLMs while employing modality perceivers to model temporal sequences. However, this approach significantly limited the model's ability to fully understand video sequences. Alternatively, some models (Liu et al., 2024e;b; Yang et al., 2025) rely on the LLM itself to model video sequences, achieving good video understanding results. Nonetheless, this method is limited to handling a small number of frames and does not support the comprehension of long videos. Understanding long videos is also a hot topic in video LLMs Jin et al. (2023); Li et al. (2023d); Zhang et al. (2024b); Shen et al. (2024); Liu et al. (2024c); Xu et al. (2024b;a); Zohar et al. (2024); Liu et al. (2024g); Shu et al. (2025); Zhang et al. (2025); Qu et al. (2024). MovieChat (Song et al., 2024) and Flash-VStream (Zhang et al., 2024a) use memory structures to process streaming videos, while LongVA Zhang et al. (2024b) and Kangaroo Liu et al. (2024c) extend the LLM's context length to accommodate more frames. Video-XL Shu et al. (2025) leverages MLLMs' inherent KV sparsification capacity to condense the visual input. Most similar to our work, PLLaVA (Xu et al., 2024a), as well as LLaVA-Video and Qwen2.5-VL, employs the non-parametric AdaptiveAvgPool function to compress visual tokens. In contrast, our method supports not only token compression but also the extraction of visual features pertinent to user prompts. This enables our model to perform more aggressive compression while preserving performance, making it suitable for both short and long video inputs. Furthermore, our convolution-style pooling method enables flexible output sizes.

**Key Content Extraction.** Extracting key video content is also a common research topic, and these models also tend to leverage CLIP to extract semantic priors Shen et al. (2024); Wang et al. (2024b;c); Park et al. (2024). For example, VideoAgent Wang et al. (2024b) iteratively searches for key frames relevant to user instructions by utilizing both generated dense captions and CLIP similarity. In comparison, these methods resemble frame selection strategies, whereas PPLLaVA represents a complete model framework. Additionally, these models typically scale up runtime, as

searching frame by frame is computationally expensive. In contrast, a major advantage of PPLLaVA is its ability to significantly enhance the efficiency of Video LLMs.

In fact, PPLLaVA functions more similarly to a Q-Former within LLaVA, but it offers several advantages over directly training a Q-Former: (1) PPLLaVA introduces far fewer additional parameters and computational overhead, amounting to less than one-tenth of a Q-Former. (2) While a Q-Former requires a three-stage pretraining process—contrastive learning, alignment training, and instruction tuning—PPLLaVA can be utilized solely during instruction tuning, allowing for seamless transfer from the most advanced MLLMs. (3) PPLLaVA supports flexible output sizes for different modalities, whereas the number of queries in a Q-Former is fixed once set. As a result, different Q-Formers typically need to be trained separately for images and videos (Zhang et al., 2023; Li et al., 2023c; Liu et al., 2024f).

## 3 APPROACH

### 3.1 MOTIVATION AND ANALYSIS

In the previous section, we discussed that the videos are redundant in both length and content. Vista-LLaMA (Ma et al., 2024) demonstrated that the extensive number of tokens in long videos makes it difficult for LLMs to capture video content. In this section, we further examine whether redundant video content impacts the performance of video LLMs and whether extracting key video content can enhance performance. Inspired by EgoSchema (Mangalam et al., 2024), we adopt the certificate length to measure the redundancy. The certificate length of a video-QA pair is determined by the shortest video sub-clip that can answer the question. Instead of using manual annotation, we employed an automated method to determine the certificate. Specifically, frames are sampled at 2 fps, and then the similarity between each frame and the question-answer text is calculated using CLIP-L-336 (Radford et al., 2021). If the similarity exceeds 0.5, the frame is considered relevant to the text. Finally, the proportion of relevant frames to the entire video is calculated as the certificate.

Based on the Video-MME dataset, we selected the 100 video-QA pairs with the shortest certificate lengths termed Video-MME-redund. We then evaluated the performance of various models on both the full Video-MME dataset and these selected samples. Additionally, for these 100 samples, we manually selected the frames most

Table 1: The study on the impact of video redundancy, we used the Vicuna-7B version for all models. "Average" and "Manual" refer to the default average sampling and manual selection, respectively.

| Model | Frames&Tokens | full | redund | |
|---|---|---|---|---|
| | | average | average | manual |
| InstructBLIP | 32&1024 | 39.2 | 36.1 | 39.5 |
| LLaVA-Next | 32&4608 | 41.1 | 36.9 | 42.0 |
| LLaVA-Next-Video | 8&1152 | 42.9 | 39.0 | 43.5 |
| LLaVA-Next-Video | 32&4608 | 45.0 | 41.5 | 46.1 |
| PPLLaVA (ours) | 32&1024 | 49.8 | 47.6 | 50.5 |

relevant to the questions, alongside the default frame sampling method. This approach was used to test whether extracting key information enhances video understanding. As shown in Table 1, all models experienced a decline in performance on high-redundancy videos. As an earlier model, InstructBLIP performed as expected, not matching the overall performance of the more advanced LLaVA-Next. However, on high-redundancy videos, InstructBLIP, which has instruction-aware video feature extraction capabilities, declined slower than LLaVA-Next. Furthermore, when manually selected frames were used, all models showed significant performance improvements, highlighting the importance of extracting key video information for enhancing video understanding. Additionally, we clearly observed the importance of including more frames for long videos, such as those in the Video-MME dataset. These findings motivated us to explore token compression to accommodate more video frames while effectively extracting key information.

### 3.2 PPLLAVA

As shown in Fig. 2, PPLLaVA, like most video LLMs, includes a vision encoder, a mapping layer, and a LLM. It also features an additional text encoder paired with the visual encoder. Given a $T$-frame video, we first pass it through the CLIP-ViT visual encoder, obtaining the visual feature $V \in \mathbb{R}^{T \times W \times H \times D}$. This feature is then fed into the Prompt-guided Pooling module, where it is

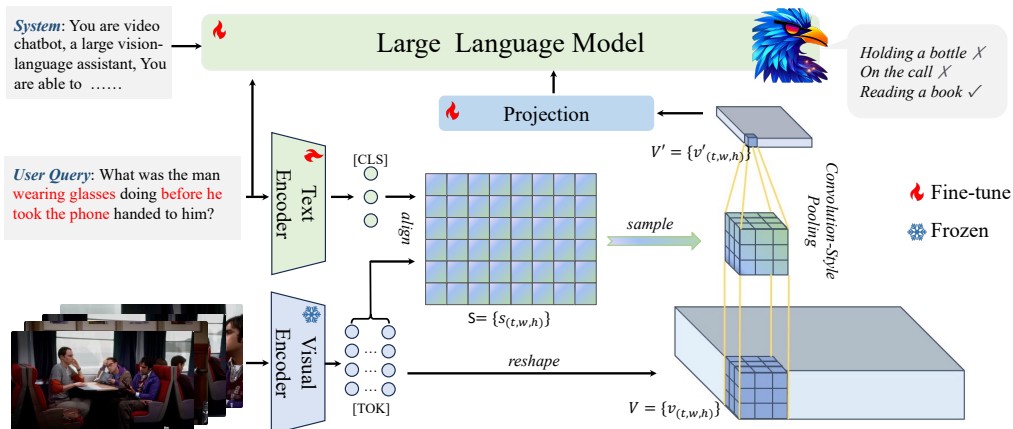

Figure 2: The overview of PPLLaVA for compressing the video based on user prompts and generating responses.

compressed by over 90%, resulting in $V' \in \mathbb{R}^{T' \times W' \times H' \times D}$. $V'$ is fed into the MLP mapping layer as the final visual input. Importantly, $V'$ not only contains significantly fewer tokens but also condenses information more relevant to the user's instructions. This ensures improved performance while efficiently processing the video input. Note that more advanced MLLMs in recent years often adopt more powerful visual encoders, such as SigLIP. However, in our model, they can be seamlessly switched. Without loss of generality, we will still refer to the visual encoder as CLIP in the following. Next, we will detail how $V'$ is obtained.

**Fine-grained Vision-Prompt Alignment.** To extract video features relevant to the prompt, we first utilize the original CLIP dual encoders to identify which video features are related to the text. Specifically, we input the user's question into the CLIP text encoder to obtain the text feature $c \in \mathbb{R}^D$. Following the CLIP training pipeline, we only use the CLS token of the text. The attention score of the $(t^{th}, w^{th}, h^{th})$ video token relative to the text feature is then calculated as:

$$s_{(t,w,h)} = \frac{\exp(\tau c \cdot f_{clipv}(v_{(t,w,h)}))}{\sum_{t=1}^{T} \sum_{w=1}^{W} \sum_{h=1}^{H} \exp(\tau c \cdot f_{clipv}(v_{(t,w,h)}))}, \tag{1}$$

where $v_{(t,w,h)}$ represents the token at the $(t, w, h)$ position in $V$, $\tau$ is the CLIP temperature scale, and $f_{clipv}$ is the CLIP visual projection, which is typically not used in multimodal LLMs. Note that $v_{(t,w,h)}$ typically refers to the patch token from the penultimate layer of CLIP, rather than the CLS token from the final layer used during CLIP training. However, since the spatial representations in CLIP's final layers are similar, applying $f_{clipv}$ still allows the patch tokens to be mapped into the interaction space with the text.

**Prompt-Guided Pooling.** In the previous section, we obtained token-level weights corresponding to the user's prompt, which we use as guidance for pooling the video. Unlike traditional tasks that require only a 1D-dimensional feature for contrastive learning (Ma et al., 2022; Wang et al., 2022), our approach aims to preserve a certain 3-dimensional structure to enable the LLM to perform temporal modeling. To achieve this, we perform pooling with $S = \{s_{(t,w,h)}\}$ in a manner similar to 3D convolution. Specifically, we define the spatiotemporal 3D convolution kernel and stride as $(k_t, k_w, k_h)$ and $(d_t, d_w, d_h)$, respectively. The output dimension of $V'$ can then be expressed as:

$$\begin{aligned}
T' &= (\frac{T - k_t}{d_t}) + 1, \\
W' &= (\frac{W - k_w}{d_w}) + 1, \\
H' &= (\frac{H - k_h}{d_h}) + 1.
\end{aligned} \tag{2}$$

Unlike conventional convolution kernels, our kernel parameters are derived from $S$. Moreover, the parameters of the kernel are dynamic; as the kernel slides over different positions in $V$, its parameters

are taken from the corresponding positions in $S$. Finally, the feature at position $(t, w, h)$ in the output $V'$ is calculated as:

$$v'_{(t,w,h)} = \sum_{i=0}^{k_t-1} \sum_{j=0}^{k_w-1} \sum_{k=0}^{k_h-1} v_{(t*d_t+i, w*d_w+j, h*d_h+k)} \cdot$$
$$s_{(t*d_t+i, w*d_w+j, h*d_h+k)} \tag{3}$$

By flexibly adjusting stride and kernel size, we can control the output dimensions. This approach allows us to better accommodate videos of varying lengths and facilitates joint training with images, compared to fixed-output methods.

**CLIP Context Extension.** In our method, CLIP-text is the only additional parameter used. Despite having significantly fewer parameters than Qformer, it achieves better performance. However, CLIP-text has a major limitation: its context length is too short (77 for CLIP and 64 for SigLIP). While this length is sufficient for objects or simple descriptions, it is inadequate for long prompts or multi-turn dialogues in multimodal LLMs. To address this performance bottleneck, we propose extending the context length of CLIP-text using asymmetric positional embedding extensions. In most cases, extending the positional embedding involves randomly initializing new embeddings at the end. A more theoretically sound approach is to perform linear interpolation on the original positional embedding at a rate of $r$. Assuming the original and target positional embeddings are $P$ and $P'$, respectively, the $i^{th}$ position of $P'$ is:

$$P'_i = P_{\lfloor j \rfloor} + (j - \lfloor j \rfloor) \cdot (P_{\lfloor j \rfloor+1} - P_{\lfloor j \rfloor}), \quad j = i \cdot r, \tag{4}$$

where $\lfloor j \rfloor$ means taking the floor of $j$. However, we found linear interpolation yielded inferior results to randomly initializing embeddings at the end. We believe this is because CLIP's positional embeddings are well-trained, and globally averaged interpolation disrupts the well-pre-trained information. Given that short sentences dominate CLIP's training data, the earlier parts of positional embeddings are more thoroughly trained. Hence, we adopted asymmetric interpolation, applying different interpolation rates at different positions. In the early part of the new positional embedding, we use a large $r$ value to shorten the interpolation distance, while in the later part, we use a smaller $r$ value to extend the interpolation distance. This asymmetric approach allows us to effectively extend the context length of CLIP-text while preserving as much pre-trained information as possible.

### 3.3 TRAINING

PPLLaVA enables plug-and-play transfer of either image-domain LLMs or image-video unified MLLMs. As a result, initialized from well-pretrained MLLMs, we can bypass expensive contrastive or alignment pretraining and proceed directly to instruction tuning. In this stage, we fully fine-tune the LLM, the projection MLP, and the CLIP text encoder. Our instruction datasets include multi-turn and single-turn conversations presented in a conversational format, along with various forms of visual input such as images, videos, and multiple images. For different types of data, we employed an interleaving training approach. Rather than using batches composed of a single data type, we mixed various data types within the same batch. This training method enables the model to simultaneously process both long videos with many frames and single-frame images, greatly enhancing its adaptability to visual sequences of varying lengths.

## 4 EXPERIMENT

In this section, we have performed comprehensive evaluations of PPLLaVA, covering settings, comparisons, and ablations, while visualizations and limitations analysis can be found in the appendix.

### 4.1 EXPERIMENT SETUP

**Implementation Details.** To verify the effectiveness of PPLLaVA across different model domains and to ensure fair comparisons with models from different periods, we implemented PPLLaVA on three separate models: the image-domain model LLaVA-Next Liu et al. (2024b), the video-domain model LLaVA-Video Zhang et al. (2024c), and general-domain VLM model InternVL3-8B (Zhu et al., 2025). Moreover, these models employ different visual encoders—CLIP, SigLIP Zhai et al.

Table 2: PPLLaVA performance on 7 video benchmarks, including NextQA, EgoChema, ActivityNet, VideoChatGPT-Bench, MVBench, LongVideoBench, and VideoMME. All results are reported as 0-shot accuracy on 7B or 8B size model.

| Models | NextQA | EgoSchema | A-Net | VCG-Bench | MVBench | L-V-Bench | VideoMME | |
| | | | | | | | Long | Overall |
| --- | --- | --- | --- | --- | --- | --- | --- | --- |
| *Open-Source Video MLLMs* | | | | | | | | |
| Video-ChatGPT (Maaz et al., 2023) | - | - | 35.2 | 2.42 | 32.7 | 39.1 | - | - |
| LLaMA-VID (Li et al., 2023d) | - | 38.5 | 47.4 | 2.89 | - | - | - | - |
| ChatUniVi (Jin et al., 2023) | - | - | 45.8 | 2.99 | - | - | 35.8 | 40.6 |
| LLaVA-NeXT-Video (Liu et al., 2024b) | 70.2 | 43.9 | 53.5 | 3.26 | - | 50.5 | - | 46.5 |
| VideoAgent Wang et al. (2024b) | 71.3 | 54.1 | - | - | - | - | - | - |
| VideoTree Wang et al. (2024c) | 75.6 | 61.1 | - | - | - | - | - | - |
| LVNet Park et al. (2024) | 72.9 | 61.1 | - | - | - | - | - | - |
| STLLM (Liu et al., 2024e) | - | - | 50.9 | 3.15 | 54.9 | - | 31.3 | 37.9 |
| VideoLLaMA2 (Cheng et al., 2024) | 75.6 | 51.7 | 53.0 | 3.12 | 57.3 | - | 43.8 | 46.6 |
| LongVA (Zhang et al., 2024b) | 69.3 | - | - | 3.19 | - | - | 46.2 | 52.6 |
| VideoChat2 (Li et al., 2023c) | - | 54.4 | 49.1 | 2.98 | 51.1 | 36.0 | 33.2 | 39.5 |
| PLLaVA (Xu et al., 2024a) | - | - | 56.3 | 3.12 | 46.6 | - | - | - |
| LLaVA-OneVision (Li et al., 2024a) | 79.4 | 60.1 | 56.6 | 3.49 | 56.7 | 56.4 | 46.7 | 58.2 |
| Video-XL (Shu et al., 2025) | - | - | - | 3.17 | 55.3 | - | - | 55.5 |
| LLaVA-Video (Zhang et al., 2024c) | 82.2 | 57.3 | 56.5 | 3.52 | 58.4 | 58.2 | 50.6 | 63.2 |
| Apollo (Zohar et al., 2024) | - | - | - | - | - | 58.5 | - | 61.3 |
| Oryx-1.5 (Liu et al., 2024g) | 81.8 | - | - | 3.62 | 57.6 | - | - | 58.8 |
| InternVL3 (Zhu et al., 2025) | - | - | - | - | 75.4 | 58.8 | - | 66.2 |
| VideoLLaMA3 (Zhang et al., 2025) | 84.5 | 63.3 | **61.3** | - | 69.7 | 59.8 | - | 66.3 |
| PPLLaVA (LLaVA-Next) | 74.9 | 60.1 | 56.1 | 3.32 | 59.2 | 53.6 | 47.4 | 53.6 |
| PPLLaVA (LLaVA-Video) | 84.1 | 61.6 | 59.7 | **3.66** | 58.8 | **60.4** | 54.3 | 64.5 |
| PPLLaVA (InternVL3) | **86.8** | **63.9** | 60.3 | 3.61 | **75.6** | 60.3 | **56.6** | **67.1** |

(2023), and InternVIT Chen et al. (2024b), respectively—which further validates the generalization ability of PPLLaVA across diverse visual features. For image and multiple-image inputs, the pooling kernel and strides are set to $(1, 3, 3)$. For video inputs, we uniformly sample 32 frames and set the pooling kernel and strides to $(2, 3, 3)$, compressing the video tokens by 18 times — a much more aggressive compression compared to the 4× reduction used in Qwen-VL or LLaVA-Video. For CLIP context extension, when $i < 20$, $r$ is set to 1, and when $i \geq 20$, $r$ is set to 0.25. We train for one epoch using a learning rate of $2e - 5$ and a batch size of 256. Since InternVL3-8B employs InternViT-300M, which undergoes extensive post-training after the initial contrastive pre-training, we re-conducted contrastive pre-training to ensure alignment between the visual and text encoders. Specifically, we used CLIP-L14-text (with InternViT-300M initialized from CLIP-L14) together with the frozen InternViT-300M, trained on 10M image–text pairs sampled from LAION Schuhmann et al. (2021) and Wukong Gu et al. (2022). The post-training aligned CLIP-L14-text was then adopted as the initialization of PPLLaVA-InternVL3. The full training takes 36 hours on 16 A100 GPUs or 32 910B NPUs.

**Data Details.** The instruction tuning data includes diverse modalities and sources. We randomly sampled 300k image data from the LLAVA-1.5 training set (Liu et al., 2024a) and used 594k multiple-image data from LLAVA-Interleave (Li et al., 2024b). The video data includes Kinetics (Kay et al., 2017), SthSth-V2 (Goyal et al., 2017), Next-QA (Xiao et al., 2021b), CLEVRER (Yi et al., 2019), and LLAVA-Video-300k, resulting in a total of 1.16M multimodal training samples.

We evaluate our model on 7 video LLM benchmarks, categorized into two types based on the evaluation method: GPT-based evaluation and multiple-choice questions. The GPT evaluation mainly involves open-ended QA, including the VCG Bench (Maaz et al., 2023) and ActivityQA (Caba Heilbron et al., 2015). Consistent with most models, we used the GPT-3.5-turbo-0613 version for testing. The multiple-choice question benchmarks include NextQA Xiao et al. (2021a), EgoSchema Mangalam et al. (2024), MVBench (Li et al., 2023c), LongVideoBench (Wu et al., 2024) and VideoMME (Fu et al., 2024). For medium-to-long videos in Video-MME and LongVideoBench, we sampled 64 frames instead of the 32 frames used in other datasets. Our test corpus encompasses videos of various genres and lengths, offering a comprehensive evaluation of PPLLaVA's performance.

Table 3: The ablation study on model components. TP means throughput (seconds/video).

| Model | Context Length | VCG Bench | | | | | | | Video-MME (w/ subs) | | | | |
|---|---|---|---|---|---|---|---|---|---|---|---|---|---|
| | | CI | DO | CU | TU | CO | Avg | TP | Short | Medium | Long | Overall | TP |
| LLaVA-Next (Average Pooing) | 576 | 3.05 | 3.07 | 3.71 | 2.62 | 3.01 | 3.09 | 2.9 | 53.1 | 41.3 | 36.0 | 43.4 | 3.1 |
| LLaVA-Next (w/o Pooing) | 4608 | 3.23 | 3.08 | 3.82 | 2.75 | 3.11 | 3.20 | 15.0 | 58.4 | 45.1 | 38.8 | 47.4 | 15.2 |
| +Prompt-guided Pooling | 1024 | 3.21 | 3.15 | 3.80 | 2.88 | 3.02 | 3.21 | 4.6 | 59.0 | 45.6 | 42.2 | 48.9 | 5.3 |
| +CLIP Context Extension | 1024 | 3.32 | 3.20 | 3.88 | 3.00 | 3.20 | 3.32 | 4.6 | 59.7 | 48.6 | 44.0 | 50.0 | 5.3 |

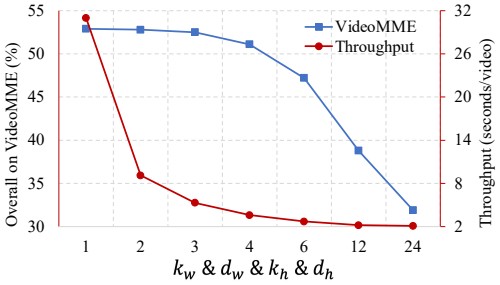 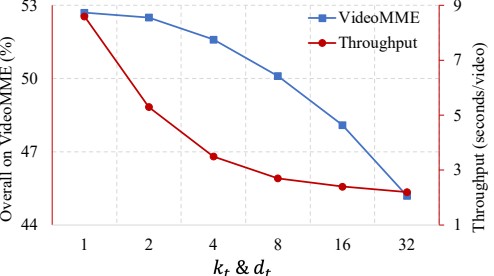

Figure 3: Spatial pooling effects. We set $T = 16$ and $k_t = d_t = 1$, varying the spatial kernel size and stride.

Figure 4: Temporal pooling effects. We set $T = 32$ and $k_w = d_w = k_h = d_h = 3$, varying the temporal kernel size and stride.

## 4.2 MAIN RESULT

Table 2 provides a quantitative comparison across several video understanding benchmarks. The results indicate that PPLLaVA consistently surpasses previous state-of-the-art models on evaluated datasets. For example, compared with LLaVA-OneVision, PPLLaVA-LLaVAVideo achieves performance gains of 4.7%, 1.5%, 3.1%, 4%, and 6.3% on NextQA, EgoSchema, ActivityNet, LongVideoBench, and VideoMME, respectively. Importantly, PPLLaVA excels in handling long video content. Under similar frame sampling settings but with significantly fewer tokens, it outperforms LLaVA-Video and LLaVA-OneVision by 3.7% and 7.6%, respectively, on VideoMME videos longer than 30 minutes. Compared with InternVL3, PPLLaVA also achieves a 1.6% improvement on LongVideoBench. This underscores PPLLaVA's strength in efficiently extracting essential information from highly redundant video streams within a limited context window. On benchmarks focused on reasoning, such as NextQA and EgoSchema—where the videos are relatively short—PPLLaVA still delivers strong results. This suggests that the model's ability to identify and utilize key information plays a crucial role in enhancing video reasoning capabilities. Even on VCG-Bench, where summarization and caption-type questions dominate, PPLLaVA still achieves leading results. This indicates that the improvements of PPLLaVA are not merely attributable to the semantic alignment provided by CLIP-like encoders; rather, the model has learned to adaptively extract critical video features, enabling it to perform well even when user queries contain limited information. Moreover, when compared to other models designed for key content extraction, including VideoAgent, VideoTree, and LVNet, PPLLaVA continues to maintain a clear performance lead. This demonstrates the effectiveness of incorporating motion priors in improving video understanding across a range of benchmarks. Finally, whether based on an image model (LLaVA-Next), a video model (LLaVA-Video), or a general-domain model(InternVL), PPLLaVA consistently outperforms its competitors under a similar baseline, especially considering that it also reduces the visual token length by several times. This demonstrates the generalizability of the PPLLaVA architecture, which can be seamlessly integrated into various types of VLMs.

## 4.3 ABLATIONS AND ANALYSIS

Unless otherwise stated, in the ablation studies, for efficiency, PPLLaVA is uniformly based on the LLaVA-Next version, and the training data includes all video data, with single-image and multi-image data excluded.

**Model Components.** The core of PPLLaVA is its prompt-guided token compression. To assess the impact of this feature, we conducted ablation experiments on overall components. As shown in Table 3, while the LLaVA-Next Baseline's direct averaging method is the most efficient, its performance is subpar. Directly feeding all tokens into the LLM yields reasonable results but suffers

Table 4: The image results. ⋆ means self-implementation.

| Model | Resolution | MMMU(val) | MathVista | MMB-ENG | MMB-CN | MM-Vet | SEED-IMG | MME | POPE |
|---|---|---|---|---|---|---|---|---|---|
| LLaVA-1.5-13B | 336*336 | 36.4 | 27.6 | 67.8 | **63.3** | 36.3 | 68.2 | 1531/295 | 85.93 |
| LLaVA-Next-7B | 672*672 | 35.8 | 34.6 | 67.4 | 60.6 | 43.9 | 70.2 | 1519/332 | 86.53 |
| VideoLLaVA | 336*336 | - | - | 60.9 | - | 32.0 | - | - | 84.40 |
| Chat-Univ-1.5 | 336*336 | - | - | 62.7 | - | 28.3 | - | - | 85.40 |
| LLaVA-Next-Video ⋆ | 336*336 | 34.2 | 28.9 | 64.7 | 56.7 | 44.0 | 64.6 | 1501/**351** | 83.10 |
| PPLLaVA | 336*336 | **37.9** | **34.6** | **68.9** | 62.0 | **44.7** | **70.7** | **1539**/277 | **88.46** |

from low throughput. Our Pooling module substantially improves both efficiency and performance. Extending the CLIP context further enhances results, particularly in long video understanding. The improvement in both efficiency and effectiveness underscores the superiority of our model.

**Pooling Size.** PPLLaVA can flexibly implement pooling at any scale. However, as the pooling kernel and stride increase, while efficiency improves, there will inevitably be performance degradation. Therefore, it's crucial to find balances on efficiency and performance. As illustrated in Fig. 3, we first explore the impact of pooling in the spatial dimension. It is evident that when the pooling kernel and stride are small, increasing them significantly improves efficiency, and thanks to the prompt-guided approach, the performance remains almost unaffected. In contrast, as shown in Fig. 4, pooling in the temporal dimension yields smaller efficiency gains compared to spatial scaling, with more noticeable performance degradation as the kernel and stride sizes increase. When the pooling kernel and stride are large, the efficiency gains tend to plateau, but the decline in effectiveness becomes significantly pronounced. Considering all factors, for video input, we ultimately selected a pooling kernel and stride of (2, 3, 3) to ensure a substantial improvement in efficiency while maintaining stable performance.

**Image Performance.** The PPLLaVA method can also be seamlessly applied to images. Although images do not have the same need for token compression as videos, the guidance from user prompts can still similarly enhance performance. In Table 4, we present PPLLaVA's results on various popular image LLM benchmarks. Since PPLLaVA was trained on LLaVA-1.5 image data based on LLaVA-Next, we compared the results of these two models. We also compare the image performance with LLaVA-Next-Video and other image-video unified models. As shown, PPLLaVA shows a significant advantage in image performance compared to video models, indicating that PPLLaVA has effectively retained pre-trained knowledge. Compared to image models, PPLLaVA, as a video model, still achieved better results on most benchmarks. Notably, our pooling method reduced the visual tokens to one-ninth of the original count at the same resolution. This demonstrates that PPLLaVA can achieve both performance and efficiency improvements even on image-based tasks, highlighting its potential for lightweight multimodal LLM.

**Pooling Approach.** Beyond the weighted average pooling detailed in the main text, we experimented with several alternative pooling methods guided by the prompt. First, we applied separate spatiotemporal pooling, conducting pooling operations independently on the temporal and spatial dimensions before concatenation. We also explored combinations of different pooling sizes to assess their impact. Lastly, we implemented max pooling using weights derived from the prompt as guidance.

Table 5: The ablation study on the Pooling Approach. We report the overall performance of VideoMME (w/ subs).

| Pooling Method | kernel1 | kernel2 | tokens | Overall |
|---|---|---|---|---|
| weighted average | (2,3,3) | - | 1024 | **53.6** |
| separate S-T | - | - | 608 | 44.1 |
| max pooling | (2,3,3) | - | 1024 | 52.0 |
| multiple | (1,6,6) | (8,2,2) | 1088 | 52.8 |
| multiple | (4,3,3) | (2,4,4) | 1088 | 53.2 |
| Token Merging | - | - | 2048 | 51.9 |

As shown in Table 5, spatiotemporal separate pooling demonstrates the worst performance, underscoring the importance of maintaining the 3-dimensional spatiotemporal structure during pooling. Max pooling, though slightly better, still falls short, suggesting that a few prominent features are insufficient to represent the entirety of the video. The combination of various pooling kernels performs similarly to direct weighted averaging when the context length is comparable. Furthermore, we experimented with a TOME-like (Bolya et al., 2022) token merging approach, which adaptively fuses visual tokens based on their N×N similarity. However, as shown at the bottom of Table 5, TOME not only achieves a lower compression ratio compared to PPLLaVA but also performs worse in terms of effectiveness. Consequently, we opted for weighted averaging PPLLaVA, as it provides optimal results while maintaining a simpler structure.

## 5 CONCLUSION

In this paper, we propose Prompt-guided Pooling LLaVA (PPLLaVA), a novel pooling method that simultaneously achieves token compression and prompt-aware feature extraction. While recent video LLMs can handle long video inputs via extended context lengths, they often suffer from excessive visual tokens, resulting in high computational costs and limited scalability. To address this efficiency bottleneck, our model introduces three key components: Fine-grained Vision-Prompt Alignment, Prompt-Guided Convolution-Style Pooling, and CLIP Context Extension. These modules enable aggressive reduction of visual context while preserving instruction-relevant information. Extensive experiments demonstrate the effectiveness of PPLLaVA across a wide range of tasks and video lengths, achieving state-of-the-art performance with significantly improved efficiency—particularly excelling in long video understanding while maintaining strong results on short video and image tasks.

**Acknowledgements.** This work was supported by National Science and Technology Major Project (2024ZD01NL00101) and National Natural Science Foundation of China (62302227).

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

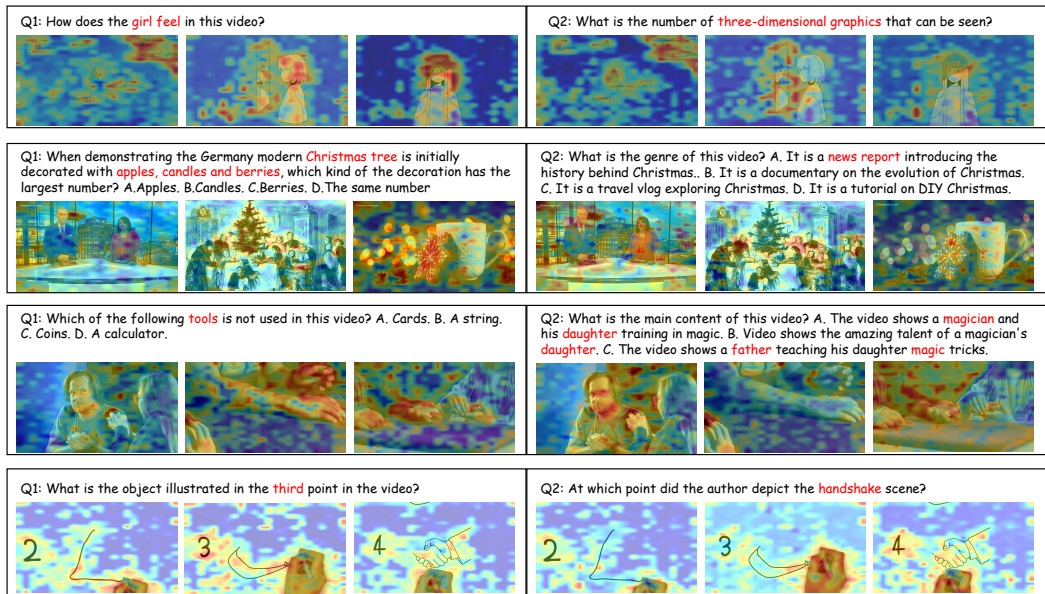

Figure 5: The visualization of the attention weights used to guide video pooling.

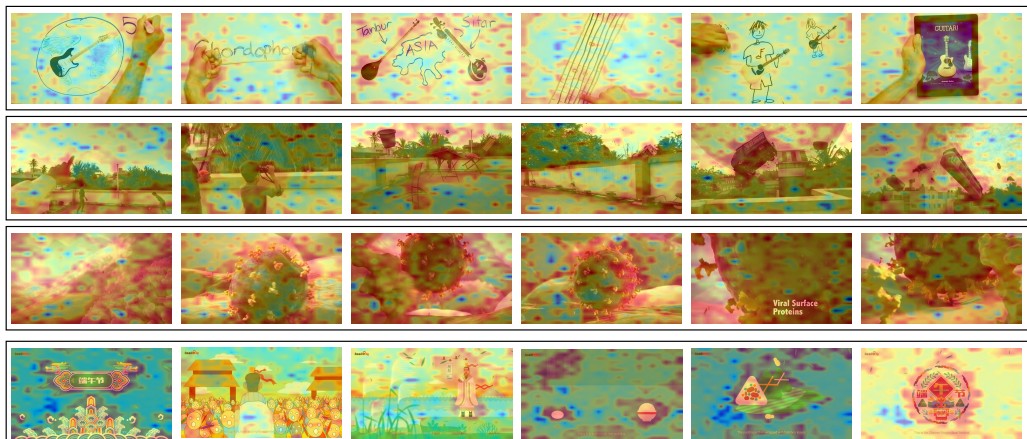

Figure 6: The visualization of the attention weights used to guide video pooling. The prompts for all videos are "Describe this video.".

## A QUALITATIVE RESULTS

In Fig. 5, we visualize the attention weights used to guide video pooling based on the user prompts. For the same video, we tried different questions. It can be clearly observed that the model's attention shifts noticeably depending on the question. For example, when the user asks about the girl's feelings, the attention is significantly focused on her face. Conversely, when asked about the number of 3D objects in the video, the attention shifts more toward the 3D objects. These visualizations demonstrate that while reducing the visual context, PPLLaVA effectively captures the key information in the video. In Fig. 6, we additionally illustrate the attention weights for captioning-related questions, as these questions theoretically provide less informational content. As shown in the figure, prompts like "Describe this video," which lack specific references, result in attention weights being evenly distributed across the foreground. This indicates that our model still plays a significant

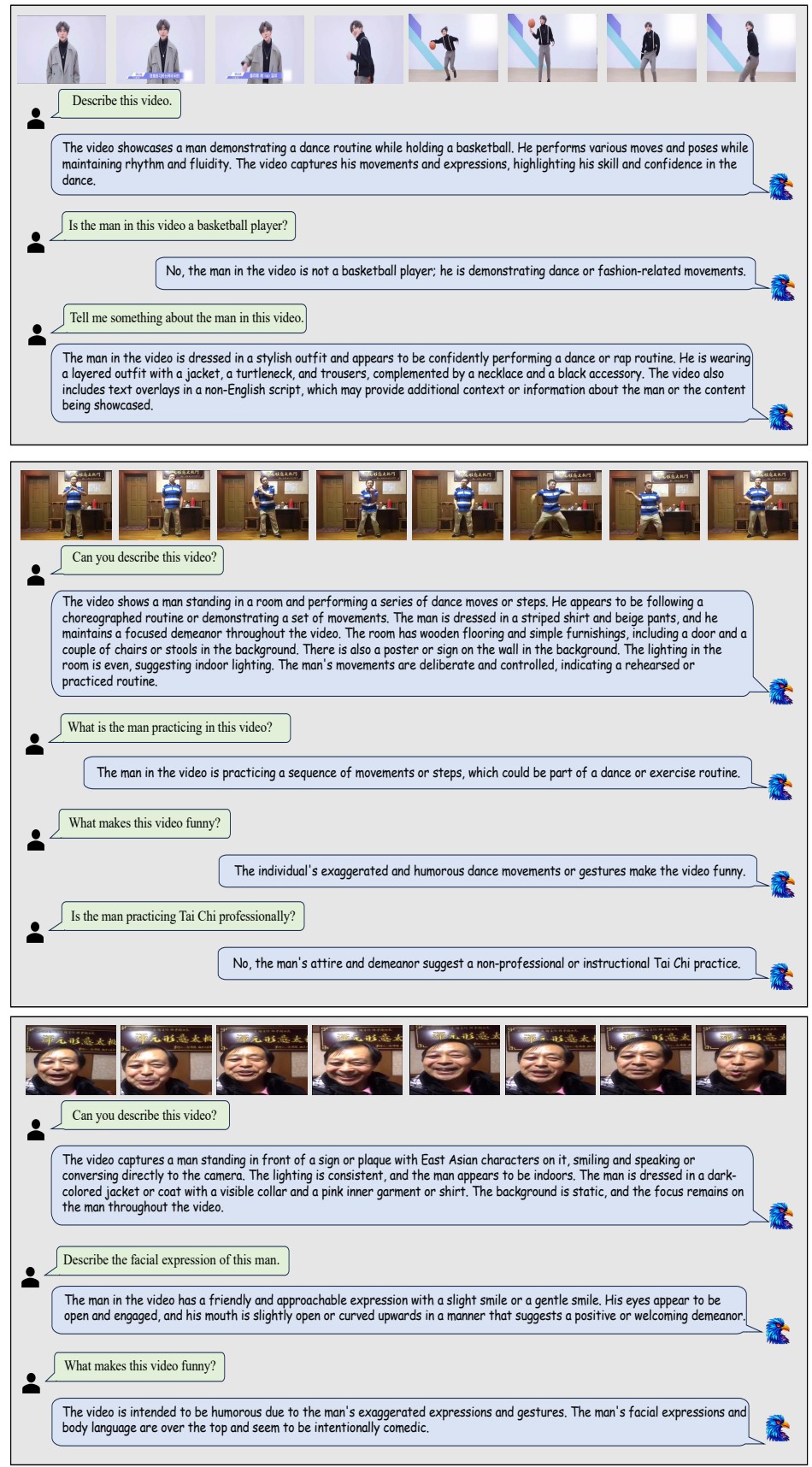

Figure 7: Qualitative result of multi-turn video conversation and reasoning.

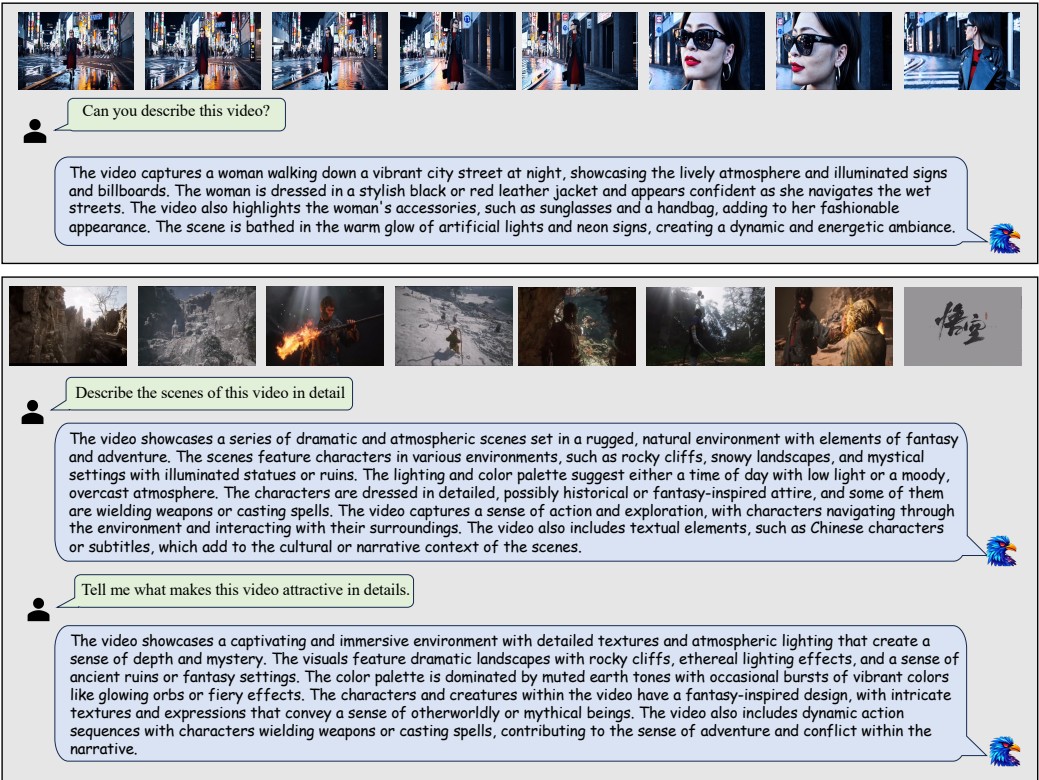

Figure 8: Qualitative result of video summary and detailed video description.

role in handling captioning-related questions. In Fig. 8 and 7, we further present some examples of video dialogue. As shown in Fig. 8, for the famous Sora video, PPLLaVA can accurately and intricately describe details about the protagonist and the environment. For the more complex scene changes in the trailer for Black Myth Wu Kong, PPLLaVA remarkably captures the details of each scene and character. In Fig. 7, PPLLaVA maintains accuracy and consistency across multiple rounds of dialogue and is capable of making reasonable inferences on open-ended questions.

## B  LIMITATION

**Limited model size and context.** Although the 7B PPLLaVA has demonstrated impressive performance, even rivaling that of 34B video LLMs, our biggest regret is that, due to having only eight A100 GPUs, we were unable to train a larger model or extend the context length to uncover the full potential of this architecture. Current state-of-the-art MLLMs, such as LLaVA-OneVision, typically adopt an 8K or longer context, which requires at least 64 A100 GPUs for training—an expense we cannot afford. On the other hand, the smaller context length ensures PPLLaVA's exceptional efficiency, representing a trade-off between efficiency and performance.

**Dependency on user prompt.** PPLLaVA utilizes user instructions to compress visual tokens and extract relevant information. However, when the user instruction carries limited information (e.g., "Describe this video"), it may impact the model's performance. In fact, this is a common limitation of MLLM methods that rely on semantic priors Park et al. (2024); Wang et al. (2024b); Shen et al. (2024); Wang et al. (2024c;a). Nonetheless, there is evidence that PPLLaVA can partially mitigate the impact of insufficient information in user prompts: (1) Theoretically, numerous studies have shown that Q-former can adaptively extract key video features, even when given minimal or no user prompts. Since PPLLaVA functions as a more lightweight and flexible version of Q-former, it may also inherit Q-former's ability to adaptively capture essential video features. (2) Quantitatively, in the VideoChatGPT-Bench, over 50% of the questions are caption-style prompts such as "What is happening in the video?" and "What is the sequence of events in the video?" Despite this, our method still performs well on VideoChatGPT. (3) Qualitatively, in Fig. 6, we visualize the attention weights

used to guide video pooling for captioning-related questions. As shown in the figure, despite the lack of sufficient information in prompts like "Describe this video," the attention weights still manage to distribute relatively evenly across the foreground while filtering out some meaningless background. This suggests that our model remains effective in handling captioning-related questions.

## C  USE OF LLM

In this work, the LLM is used solely for language polishing and serves no other purpose.

