# OpenReview forum: "PPLLaVA: Varied Video Sequence Understanding With Prompt Guidance"
_ICLR.cc/2026/Conference — ICLR 2026 Poster_

### Official Review · Reviewer_Qq7T · 2025-10-20

**Soundness:** 3
**Presentation:** 3
**Contribution:** 3
**Rating:** 6
**Confidence:** 3

**Summary:**

This paper presents PPLLaVA, a prompt-guided pooling framework designed for video-based large language models (Video LLMs). PPLLaVA focuses on compressing redundant visual tokens in video sequences while maintaining semantics that are critical to user instructions. The method achieves this through fine-grained vision–prompt alignment and a novel convolution-style pooling mechanism guided by textual prompts. In addition, it extends CLIP’s textual context to better support complex, multi-turn visual dialogues. Extensive experiments demonstrate that PPLLaVA attains state-of-the-art accuracy and efficiency across a broad range of image and video understanding benchmarks, delivering notable gains in both computation speed and task performance—particularly for long-duration video analysis.

**Strengths:**

**1. Validity and Soundness**:
PPLLaVA introduces a highly effective token compression strategy, achieving a reduction ratio of up to 18x. This aggressive compression directly confronts a significant bottleneck in Video LLMs: the prohibitive computational cost associated with long context lengths. Notably, this reduction does not compromise, and in some cases even enhances, performance compared to state-of-the-art models across diverse video understanding benchmarks (Table 2).

**2. Strong Generalization**:
The model's robust generalization is further highlighted by the experiments in Table 2. These results show that PPLLaVA can be seamlessly integrated with various mainstream MLLM backbones (such as LLaVA-Next, LLaVA-Video, and InternVL3), yielding consistent performance improvements across the board. This adaptability underscores its powerful generalization and transfer capabilities.

**3. Clarity and Concrete:**
The paper is characterized by its clarity. The mathematical exposition is particularly lucid, especially the detailed formulations governing the alignment and pooling mechanisms (see equations in Section 3.2). Furthermore, the end-to-end pipeline is effectively illustrated in Figure 2, providing a clear visual representation of the entire architecture.

**Weaknesses:**

**1. CLIP Context Extension Ablation:**
 The paper posits the specific contribution of the CLIP context extension (asymmetric positional embedding). However, isolated ablation results for this module are not provided in Table 3. It would be valuable to evaluate its effect in isolation from the prompt-guided pooling module and to further compare it with other methods designed to extend CLIP's text length, such as Long-CLIP[1] and VideoCLIP-XL[2].

**2. Missing Efficency Analysis:**
Efficiency claims (18× compression) are strong but lack FLOPs or latency analysis (just “throughput (s/video)” — not standardized).

**3. Potential Over-reliance on CLIP-Based Semantics:**
The method's core relies on CLIP’s ability to associate text and visual fragments. While the authors acknowledge this (pages 2–3), there is no systematic analysis of whether this dependency limits extensibility to domains beyond CLIP’s pretraining. Furthermore, the CLIP context extension is only heuristically motivated and lacks ablation results isolating this component’s pure effect.

**4. Minor Issue:**
line 021, "clip" should be "CLIP"

[1] Zhang B, Zhang P, Dong X, et al. Long-clip: Unlocking the long-text capability of clip[C]//European conference on computer vision. Cham: Springer Nature Switzerland, 2024: 310-325.

[2] Wang J, Wang C, Huang K, et al. Videoclip-xl: Advancing long description understanding for video clip models[J]. arXiv preprint arXiv:2410.00741, 2024.

**Questions:**

1.  Are the CLIP-based relevance scores $S=\{s(t,w,h)\}$ computed once (frozen) or updated during training?  If CLIP is frozen, how does the model adapt to new prompt distributions?
2. How was the interpolation rate (r=1 before 20, r=0.25 after 20) in CLIP Context Extension chosen, empirically or theoretically?
3. The paper reports “throughput (s/video)” but not FLOPs, GPU memory, or latency at batch=1. Please report actual compute savings (e.g., FLOPs, peak memory, tokens/sec) to make the 18× compression claim quantifiable.
4. How sensitive is PPLLaVA to the CLIP backbone (ViT-L/14 vs. ViT-H/14 vs. SigLIP)? Have you tested whether replacing CLIP with a weaker vision encoder (e.g., EVA-CLIP) preserves benefits?

---

> ### Author Response · Authors · 2025-11-19
> **Response to Reviewer  Qq7T (the first part)**
>
> Thanks for the valuable comments. We appreciate you giving positive feedback on our paper.  We have addressed all of your concerns point by point in two pages. It is appreciated if you have any further feedback on our response.
>
>  ### Q1: CLIP Context Extension Ablation
>  Thank you for the suggestion. In Table 3, we indeed only ablated the effect of adding the CLIP Context Extension module and did not further investigate different methods designed to extend CLIP’s text length. Here, we would first like to emphasize that, for training efficiency, we do not use additional CLIP visual–text encoders. The text encoder used for initialization must be paired with the visual encoder of the original VLM; directly using a pretrained but unpaired text encoder would clearly lead to performance degradation. To address the concern, we additionally experimented with a LongCLIP text encoder trained from scratch to evaluate its effect:
>
>  | Method   | VideoMME|LongVideoBench|
> |:--------|:------:|:------:|
> | baseline (P-Pooling w/o extension)  | 48.9 | 52.1 |
> | +Long-CLIP (Pretrained)   |45.3  | 49.9|
> | +Long-CLIP (No-Pretrain)   |49.7  | 52.6|
> | +Context Extension (Ours)   |50.0| 53.6 |
>
> As shown, the pretrained LongCLIP leads to a severe performance drop. In contrast, LongCLIP initialized directly from the original CLIP still provides a noticeable improvement, which demonstrates the importance of extending text context length in our framework. However, the gains from LongCLIP are still smaller than those brought by our Context Extension module. We hypothesize that this is because LongCLIP introduces more aggressive modifications to the text positional encoding, which typically requires substantial contrastive learning to be fully optimized. In our setting, however, the model is only supervised by the decoder loss, and to avoid degrading the VLM’s performance, the visual encoder is trained with a relatively small learning rate. This likely leads to insufficient optimization for aggressive text encoder changes. Therefore, our module is better suited for plug-and-play training scenarios in VLMs.
>
>  ### Q2: Missing Efficency Analysis
> Thank you for the suggestion. We report throughput rather than FLOPs mainly for comparability, as only a few existing works report throughput and almost none provide FLOPs. To further support our efficiency claims, we additionally compare the FLOPs of the baseline and our method:
>
>   | Method   | Frame|Tokens|TFLOPs
> |:--------|:------:|:------:|:------:|
> | baseline  | 64 | 576*64 | 1220
> | baseline (avg-pool)  | 64 | 64*32| 39.53
> | Ours  |64 | 64*32| 39.57
>
> As shown, directly feeding all visual tokens into the LLM requires up to 1220 TFLOPs, while controlling the visual sequence length reduces the cost dramatically to 39.53 TFLOPs. Our Prompt-Guided compression module introduces only an additional 0.04 TFLOPs. Overall, our method achieves over **30×** FLOPs improvement, further demonstrating its efficiency.
>
> ### Q3: Potential Over-reliance on CLIP-Based Semantics
> It is true that CLIP’s pretraining domain may lack certain capabilities, such as aligning with complex queries or understanding densely temporal scenarios. However, we would like to emphasize that both our visual and text encoders are further trained, with the text encoder in particular being optimized with a larger learning rate. In the table below, we report the performance on several sub-tasks of the benchmark, comparing the Prompt-Guided Pooling module without training (relying solely on CLIP’s initial alignment) versus the trained module:
>
> | Method   | VideoMME|VideoMME|MVBench|MVBench
> |:--------|:------:|:------:|:------:|:------:|
> |   | Temporal Perception | Temporal Reasoning | Moving Direction| Scene Transition
> | baseline  | 72.1 | 45.4 | 34.0 | 88.5
> | PPLLaVA (w/o training)  | 67.3 | 40.7| 25.5| 69.5
> | PPLLaVA (w training)  | 76.4 | 46.9 | 37.5 | 83.0
>
> As shown, for these highly temporally sensitive tasks, directly using the Prompt-Guided Pooling module leads to a severe performance drop, highlighting the limitations of CLIP’s pretraining. However, after training, performance on these sub-tasks recovers substantially and, in most cases, surpasses the baseline. From this, we conclude that while CLIP provides a strong initialization for our method, our approach is not constrained by CLIP’s pretraining domain. With training, the model can unlock and enhance capabilities that CLIP alone does not provide, such as dynamic understanding and long-context comprehension. Additionally, the ablation of the CLIP Context Extension has already been addressed in Q1.

---

> ### Author Response · Authors · 2025-11-19
> **Response to Reviewer Qq7T (the second part)**
>
> ### Q4: Are the CLIP-based relevance scores computed once (frozen) or updated during training?
>  Our CLIP-based relevance scores are of course updated during training, since the text encoder is trainable. As a result, the pooling weights vary across samples and are continuously optimized throughout training. In fact, to enable the text encoder to produce more effective pooling weights, we assign it a significantly larger learning rate compared to the LLM backbone. We will highlight this point in the implementation details.
>
>  ### Q5: How was the interpolation rate in CLIP Context Extension chosen, empirically or theoretically?
> When selecting the boundary for asymmetric interpolation, we roughly analyzed the caption lengths in COCO Caption, Flickr30k, and CC3M, and found that almost all captions are no longer than 15 words. Based on this, we manually set the boundary to 20 tokens. The choice of the interpolation rate depends on how much we want to extend the context. We analyzed the query lengths in VideoMME and LongVideoBench and found that extending the context to 256 tokens is sufficient to cover all queries. Accordingly, we set the value of r to achieve this context length.
>
>  ### Q6: Please report actual compute savings (e.g., FLOPs, peak memory, tokens/sec) to make the 18× compression claim quantifiable.
> In Q2, we have already reported the FLOPs, which are reduced by approximately 30×, even higher than the 18× token compression reported.
>
>  ### Q7: How sensitive is PPLLaVA to the CLIP backbone (ViT-L/14 vs. ViT-H/14 vs. SigLIP)? Have you tested whether replacing CLIP with a weaker vision encoder (e.g., EVA-CLIP) preserves benefits?
> In our model, the text encoder is tightly paired with the VLM’s visual encoder. For example, in the LLaVA-Next implementation, we use the CLIP-L-336 text encoder; in LLaVA-Video, we use the SigLIP text encoder; and in the InternVL experiments, we use the InternViT text encoder. Replacing the text encoder with one that is not paired with the visual encoder would significantly increase the training cost and also lead to a substantial performance drop, as the pretraining-based initialization would be disrupted. Therefore, arbitrary replacement is not feasible. Importantly, our three baseline experiments correspond to these three different CLIP variants, and all achieve strong progress. This demonstrates that PPLLaVA is not overly sensitive to the choice of CLIP backbone and exhibits strong generalization across different architectures.

---

> ### Author Response · Authors · 2025-11-27
> **Looking forward to a response**
>
> Dear Reviewer Qq7T,
>
> We want to send you a friendly reminder that it has been about nine days since our last exchange.
>
> It is greatly appreciated if you are willing to reconsider your score based on our responses, and we really want to know whether our responses address your concerns. If there is any other concern that we could not address in the response, please feel free to let us know and we would be happy to provide further explanation.
>
> With sincere regards,
>
> Authors of Submission7387

---

### Official Review · Reviewer_eVAF · 2025-10-30

**Soundness:** 3
**Presentation:** 3
**Contribution:** 3
**Rating:** 6
**Confidence:** 3

**Summary:**

This paper addresses the high computational cost and inefficiency of Video LLMs caused by processing redundant visual tokens. It proposes PPLLAVA, a model featuring a prompt-guided pooling strategy. The method uses CLIP-based visual-prompt alignment to generate a Attention map, which then functions as a dynamic 3D convolution-style kernel to aggressively compress visual tokens while retaining task-relevant information. A CLIP text context extension is also introduced to handle long prompts. Experiments on seven benchmarks demonstrate SOTA performance and significantly improved inference throughput.

**Strengths:**

1. The paper is well-motivated and addresses a critical and practical problem in Video LLM deployment: the trade-off between performance and computational efficiency. The motivation analysis in Section 3.1 (Table 1), which demonstrates performance degradation on high-redundancy videos, provides a solid empirical basis for the proposed solution.
2. The experimental validation is extensive and a significant strength of this work. The authors validate PPLLAVA's effectiveness across diverse video benchmarks.

**Weaknesses:**

1. The claim of model generality could be further substantiated. While the SOTA results on InternVL3 are strong , all in-depth ablation studies (e.g., Table 3, Table 5) are conducted exclusively on the LLaVA-Next version. Replicating key component ablations on the InternVL3 would be necessary to confirm the generalization. Furthermore, testing the method on other prominent open-source VLMs, such as the Qwen2/2.5-VL series, would significantly strengthen the paper's claims of being a general-purpose module.
2. The comparison to alternative token compression strategies is insufficient. The paper's main ablation (Table 5) only compares against internal variations (e.g., max pooling) and a "TOME" method. This TOME approach, originally proposed in 2022, is a relatively dated technique. This limited ablation does not adequately benchmark the proposed method against the current state-of-the-art. The paper would be more convincing if it included direct baseline comparisons against other full, established, and more recent token reduction methods to clearly isolate the benefits of the prompt-guided approach.

**Questions:**

See weaknesses above.

---

> ### Author Response · Authors · 2025-11-19
> **Response to Reviewer  eVAF**
>
> Thanks for the valuable comments. We appreciate you giving positive feedback on our paper. All your concerns are addressed point by point. It would be appreciated if you have any further feedback on our response.
>
>   ### Q1.1: Key component ablations on the InternVL3
> We appreciate the reviewer’s suggestion. Following the recommendation, we provide ablations on the key components of InternVL3. For all settings except the baseline, each video is sampled with 64 frames and the pooling kernel size is fixed to (2, 2, 2). The results are shown below:
>
> | Method   |tokens| VideoMME|VideoMME-Long|LongVideoBench|
> |:--------|:------:|:------:|:------:|:------:|
> | w/o pooling (baseline)  |256*32| 66.3 | 53.5 | 58.8
> | avg pooling   | 64*32 | 63.7| 52.1| 57.0
> | +Prompt-guided Pooling   |64*32|66.5  | 55.0|59.2
> | +Context Extension   |64*32|67.1| 56.6 | 60.3
>
> As illustrated, applying same-size average pooling directly leads to a substantial performance drop. In contrast, integrating Prompt-guided Pooling module yields a significant performance recovery—enabling effective token reduction while still surpassing the baseline. Furthermore, extending the context of the text encoder leads to an additional improvement, with especially notable gains on long-video benchmarks. This clearly demonstrates the generalization capability of our proposed method.
>
>    ### Q1.2: Performance on Qwen-VL series.
>    We appreciate the reviewer’s suggestion. The main reason we did not include experiments on the Qwen-VL series is that **Qwen-VL models are substantially more heavily pre-trained**. In other words, the number of training tokens used by Qwen-VL is _far larger_ than that of the InternVL series. As a consequence, continuing SFT training on Qwen-VL generally requires much more data to yield noticeable improvements (and our training set is already a subset of what Qwen-VL and InternVL have used). Unfortunately, we do not have sufficient computational resources to support such large-scale continued training.
>
>    Nevertheless, to address the reviewer’s concern, we still conducted a small experiment on Qwen2.5-VL-3B, using exactly the same training data as in our previous experiments:
>
> | Method   | VideoMME|MVBench|LongVideoBench|
> |:--------|:------:|:------:|:------:|
> | Qwen2.5-VL-3B  | 61.5 | 67.0 | 54.2
> | +PPLLaVA  | 63.0 | 68.9 | 55.3
>
> As shown, our method achieves consistent and meaningful improvements on Qwen2.5-VL as well, indicating that the proposed approach generalizes well to other mainstream MLLM architectures. With more training resources, we believe our model has strong potential to obtain even better results on larger variants.
>
>  ### Q2: The comparison to alternative token compression strategies is insufficient.
>  We believe this concern arises from a certain misunderstanding. In Table 5, we include TOME not to compare with other token compression methods, but to analyze _how the core pooling module should be designed_. This experiment serves as an architectural ablation rather than a comparison of compression algorithms.
>
>  In the main paper, we have already compared and discussed current state-of-the-art token compression approaches such as VideoAgent, VideoTree, and LongVU. Specifically, Table 2 provides a direct comparison, and the _Related Work_ section (starting at line 148) offers additional discussion. Here, we would like to reiterate the key differences and advantages of PPLLaVA over these methods:
>
>  (1) Our model is a trainable, end-to-end approach, whereas most existing token compression methods are training-free frame-selection strategies.  PPLLaVA focuses on reliably extracting query-relevant visual representations and achieving efficient compression regardless of how many frames are sampled or how they are selected. In contrast, methods such as VideoAgent and VideoTree primarily aim to identify the key frames to get lower tokens. These two directions target different objectives and are therefore complementary rather than directly comparable.
>
>  (2) Compared with other training-free token pruning approaches, our method offers significant advantages in evaluation efficiency. Methods such as VideoAgent and VideoTree require dense captioning or multiple rounds of LLM calls, while approaches like LongVU rely on per-frame feature matching and pruning, all of which introduce substantial computational overhead during inference.  In contrast, although PPLLaVA requires training, it supports direct end-to-end evaluation at test time—without additional frame selection, external captioning, or iterative feature comparison. Moreover, the extra FLOPs introduced by our lightweight compression module are negligible (see Reviewer Qq7T Q2).  Importantly, beyond the efficiency benefits, our method also outperforms these  approaches in overall effectiveness, highlighting the strength of our design.

---

> ### Author Response · Authors · 2025-11-27
> **Looking forward to a response**
>
> Dear Reviewer eVAF,
>
> We want to send you a friendly reminder that it has been about nine days since our last exchange.
>
> It is greatly appreciated if you are willing to reconsider your score based on our responses, and we really want to know whether our responses address your concerns. If there is any other concern that we could not address in the response, please feel free to let us know and we would be happy to provide further explanation.
>
> With sincere regards,
>
> Authors of Submission7387

---

### Official Review · Reviewer_ddaz · 2025-10-30

**Soundness:** 2
**Presentation:** 2
**Contribution:** 2
**Rating:** 4
**Confidence:** 4

**Summary:**

This paper proposes PPLLaVA, a video LLM that introduces a prompt-guided pooling strategy for adaptive token compression, aiming to achieve unified and state-of-the-art performance on both short and long video understanding benchmarks.

**Strengths:**

- The identification of redundant content as the core issue and the proposed prompt-guided pooling as a solution are well-motivated contributions to the field.

- The paper is well-written and easy to follow.

**Weaknesses:**

1.  The main contribution of this work is the design of a Q-Former-like module, integrated into a pre-trained MLLM, to reduce the number of visual tokens. The authors use the CLS token from the text encoder and compute its similarity with all visual tokens to parameterize a 3D convolution operation. However, since textual queries may involve spatio-temporal information, computing similarity with individual visual tokens, which lack temporal context, could introduce non-negligible errors.

2.  In the experiments, the authors continually fine-tune their method on three different MLLMs and compare it against their base versions. This comparison appears unfair, as their model has been exposed to additional training data. A more equitable evaluation would involve fine-tuning the baseline models on the same dataset for a valid comparison.

3.  The authors omit discussion and comparison with existing state-of-the-art training-free token pruning methods. When considering such approaches, the proposed method does not appear to offer clear advantages.

**Questions:**

See Weaknesses

---

> ### Author Response · Authors · 2025-11-19
> **Response to Reviewer  ddaz**
>
> Thank you very much for the valuable comments. We are sorry for some misunderstandings on our paper. We believe that our response can help clarify and address some potential misunderstandings. We have addressed all of your concerns point by point. It is appreciated if you have any further feedback on our response.
>  ### Q1: Computing similarity with individual visual tokens, which lack temporal context, could introduce non-negligible errors.
> We appreciate the reviewer’s thoughtful comment. However, this concern may stem from a misunderstanding of certain details:
>
> (1) **Visual tokens in our model are _not_ isolated per-frame patches without temporal context:**  Before computing similarity with the CLS token, the visual tokens are processed by a vision encoder that has already been _pre-trained_ to encode cross-frame temporal information. For example, the visual encoder in InternVL3 is jointly pre-trained on both images and videos with dense supervision, enabling individual visual tokens to inherently contain temporal cues, rather than representing purely frame-local features.
>
> (2) **During training, the alignment between visual tokens and temporally informed textual tokens is continuously optimized:**  In PPLLaVA, the compression kernel is 3D, explicitly incorporating the temporal dimension. At the same time, the text encoder is trained jointly with the MLLM backbone using a relatively large learning rate, which encourages it to encode temporal cues relevant to the downstream task. As a result, the alignment between visual and textual representations is optimized to capture events, actions, and temporal relations described in the query, rather than being restricted to static frame-level semantics.
>
> (3) **Empirically, if temporal misalignment were an issue, we would expect performance degradation on datasets requiring long-range temporal reasoning. However, this is not observed:**  Across long-video benchmarks such as LongVideoBench and VideoMME-Long as well as MVBench, which contains a large number of temporally sensitive queries (e.g., object motion direction and state transitions), our model consistently achieves strong performance. These results demonstrate that our method captures temporal structure sufficiently well in practice, further indicating that the proposed pooling mechanism does not suffer from the temporal misalignment.
>
> ### Q2: Unfair comparison with baseline.
> We appreciate the reviewer’s concern. However, we would like to clarify that, in terms of training data, our method does _not_ use any “additional training data” beyond the baselines—in fact, we are at a disadvantage. Both LLaVA-Video and InternVL3 are trained with substantial GPU resources on massive-scale datasets, whereas we do not have comparable computational resources. As a result, the training data used in our work is actually a **subset** of what the baseline models were trained on. Therefore, the comparison is not unfair; if anything, performing better with _less_ training data further highlights the effectiveness and efficiency of our proposed model.
>
>  ### Q3: Missing discussion and comparison with existing state-of-the-art training-free token pruning methods.
>   First, we would like to emphasize that _we have already compared and discussed_ existing sota training-free token pruning methods such as VideoAgent, VideoTree and LongVU in the main paper. Specifically, Table 2 presents a direct comparison with these two methods, and the _Related Work_ section (starting at line 148) also includes a discussion about them.
>
>  Second, compared with these approaches, our proposed method differs in important ways and offers several advantages:
>
>  (1) Our method is a trainable, end-to-end model, whereas many training-free approaches—such as VideoAgent and VideoTree—are essentially frame selection strategies. Our model focuses on reliably extracting query-relevant features and achieving efficient compression regardless of how many frames are sampled or how they are selected, while these training-free methods focus on selecting the key frames. These two directions are fundamentally different and can even be complementary.
>
> (2) In terms of evaluation efficiency, our approach has a significant advantage over training-free token pruning methods.
> Methods like VideoAgent and VideoTree require dense captioning or multiple rounds of LLM calls, while LongVU-style pruning requires per-frame feature matching and pruning. In contrast, although PPLLaVA requires training, it enables **direct end-to-end evaluation** at test time, without spending additional time on frame selection or token-level comparison. Moreover, the FLOPs introduced by our lightweight compression module are negligible (see Reviewer Qq7T Q2), making inference highly efficient. At the same time, our method also achieves superior performance compared with these training-free baselines, further highlighting its effectiveness.

---

> ### Author Response · Authors · 2025-11-27
> **Looking forward to a response**
>
> Dear Reviewer ddaz,
>
> We want to send you a friendly reminder that it has been about nine days since our last exchange.
>
> It is greatly appreciated if you are willing to reconsider your score based on our responses, and we really want to know whether our responses address your concerns. If there is any other concern that we could not address in the response, please feel free to let us know and we would be happy to provide further explanation.
>
> With sincere regards,
>
> Authors of Submission7387

---

### Official Review · Reviewer_u53s · 2025-10-31

**Soundness:** 3
**Presentation:** 3
**Contribution:** 2
**Rating:** 4
**Confidence:** 4

**Summary:**

This paper introduces PPLLaVA, a novel video understanding framework that improves both efficiency and performance by compressing visual tokens based on user prompts. The core idea is to reduce video redundancy using a prompt-guided 3D pooling mechanism, which dynamically compresses visual tokens while preserving instruction-relevant information. The model includes three key components: a CLIP-based vision-prompt alignment module to identify relevant visual regions; a prompt-guided pooling mechanism that performs 3D convolution-style pooling using prompt-based attention weights; a CLIP context extension module to support longer text inputs for multi-turn dialogues.
PPLLaVA achieves up to 18× token compression, and consistently outperforms strong baselines (e.g., LLaVA-OneVision, InternVL3) across 7 video understanding benchmarks, including long-form video tasks. It is also plug-and-play, transferable across different base models (LLaVA-Next, InternVL3, etc.).

**Strengths:**

1. Innovative and Practical Solution: The prompt-guided pooling mechanism effectively tackles visual token redundancy in long video understanding, also enhancing performance in image-only tasks, indicating broad applicability.
2. Strong Empirical Performance: PPLLaVA achieves state-of-the-art results across multiple benchmarks (e.g., Video-MME, NextQA, EgoSchema), often using fewer tokens and enabling faster inference.
3. High Generalizability and Efficiency: The method demonstrates strong transferability across various base models (image-only, video-only, and unified VLMs), significantly improving throughput—up to 3× faster—with minimal to no accuracy loss.

**Weaknesses:**

1. The prompt-guided pooling weights, derived directly from CLIP similarity scores without learnable parameters or adaptive gating, may restrict expressiveness and adaptability.
2. The paper lacks direct comparisons with other prompt-aware compression methods (e.g., VideoAgent, VideoTree) in terms of accuracy and efficiency, despite numerous SOTA model comparisons.

**Questions:**

1. Have you considered making the prompt-guided pooling weights learnable (e.g., via lightweight attention or gating mechanisms) instead of directly using CLIP similarity?
2. How does the model perform when user prompts are vague or generic (e.g., “What is happening?”)? Is there a quantitative analysis of performance degradation in such cases?

---

> ### Author Response · Authors · 2025-11-19
> **Response to Reviewer  U53s (the first part)**
>
> Thank you very much for the valuable comments. We are sorry for some misunderstandings on our paper. We believe that our response can help clarify and address some potential misunderstandings. We have addressed all of your concerns point by point in two pages. It is appreciated if you have any further feedback on our response.
> ### Q1: The prompt-guided pooling weights without learnable parameters or adaptive gating.
> We believe this concern arises from a partial misunderstanding. While the prompt-guided pooling weights themselves do not contain learnable parameters, they are generated by a learnable text encoder. As a result, the pooling weights vary across samples and are continuously optimized during training. In fact, to enable the text encoder to produce more effective pooling weights, we assign it a significantly larger learning rate compared to the LLM backbone.
>
> From another perspective, the generation of prompt-guided pooling weights can also be viewed as incorporating a gating-like mechanism: the parameters of the text encoder function analogously to those in a GLU, extracting weighted signals from visual features and applying these weights back onto the visual representations. The key difference is that our text encoder benefits from strong pre-trained alignment, which allows it to perform this gating-like operation effectively without requiring heavy additional pretraining.
>
> To further address the reviewer’s concern, we conducted a tiny experiment by introducing a _true_ GLU-style gating mechanism. Specifically, we used learnable parameters to generate weights from the visual features, applied a tanh activation, and then multiplied these weights with the visual features. The initial weights were set to be completely uniform. All other settings followed the same configuration as our ablation studies.
>
> | Method   | VideoMME|MVBench|
> |:--------|:------:|:------:|
> | baseline   | 47.4 | 53.1 |
> | GLU   | 45.9 | 53.5 |
> | PPLLaVA   |53.6| 59.2 |
> | PPLLaVA+GLU   |50.1| 58.4 |
>
> As shown in the results, the model using only GLU exhibited a significant performance drop. Moreover, combining PPLLaVA with GLU also degraded performance compared to using PPLLaVA alone. This demonstrates both the effectiveness of PPLLaVA relative to GLU and the fact that PPLLaVA already achieves a gating-like effect to some extent. At the same time, we believe this result does not imply that GLU is inherently poor; rather, compared to PPLLaVA, GLU requires substantially _heavier pretraining_ to yield observable gains.
>
>  ### Q2: Lack of direct comparisons with other prompt-aware compression methods.
>  First, we would like to emphasize that _we have already compared and discussed_ prompt-aware compression methods such as VideoAgent and VideoTree in the main paper. Specifically, Table 2 presents a direct comparison with these two methods, and the _Related Work_ section (starting at line 148) also includes a detailed discussion of them.
>
> Second, we would like to reiterate the fundamental difference between PPLLaVA and prompt-aware compression methods such as VideoAgent and VideoTree. PPLLaVA is a _trainable, end-to-end model_, whereas VideoAgent and VideoTree are _training-free frame selection strategies_. PPLLaVA focuses on extracting query-relevant features and achieving efficient compression regardless of how many frames are sampled or how they are selected; in contrast, VideoAgent and VideoTree focus specifically on how to choose key frames.
>
> In terms of effectiveness, Table 2 already demonstrates a clear advantage for PPLLaVA. Although PPLLaVA is a trained model while VideoAgent and VideoTree are training-free, they rely on external APIs such as GPT, making it difficult to guarantee a fair comparison.
> In terms of efficiency, VideoAgent and VideoTree only report the number of selected frames and LLM calls, without providing throughput or FLOPs, making a direct efficiency comparison impossible. By contrast, PPLLaVA can be evaluated end-to-end at test time and achieves significant FLOPs reduction (see Reviewer Qq7T’s Q2). Meanwhile, VideoAgent and VideoTree require multiple LLM calls (at least 2–3 rounds) or dense captioning. Therefore, under the same number of frames, PPLLaVA can offers better efficiency.

---

> ### Author Response · Authors · 2025-11-19
> **Response to Reviewer U53s (the second part)**
>
> ### Q3:  Have you considered making the prompt-guided pooling weights learnable instead of directly using CLIP similarity?
> Firstly, as explained in Q1, although PPLLaVA initializes the pooling weights using CLIP similarity, the entire module is fully learnable, and the prompt-guided pooling weights are continuously optimized during training. Secondly, as also discussed in Q1, we implemented an additional experiment where we used a GLU-based learnable gating mechanism to generate pooling weights. The results clearly show that this GLU-based design underperforms, highlighting the effectiveness and superiority of the PPLLaVA architecture.
>
>  ### Q4:  How does the model perform when user prompts are vague or generic (e.g., “What is happening?”)?
>  The issue of vague or generic prompts has been discussed in multiple places in the main paper (lines 107, 754, and 854). Qualitatively, Figure 6 shows that under vague or generic prompts, the attention weights tend to become more evenly distributed and spread across the foreground while suppressing meaningless background regions. Quantitatively, on VCG-Bench—where vague or generic prompts account for a large proportion—PPLLaVA still achieves strong performance. For more detailed analysis, please refer to the “Dependency on user prompt” section in the original paper (line 854).

---

> ### Author Response · Authors · 2025-11-27
> **Looking forward to a response**
>
> Dear Reviewer u53s,
>
> We want to send you a friendly reminder that it has been about nine days since our last exchange.
>
> It is greatly appreciated if you are willing to reconsider your score based on our responses, and we really want to know whether our responses address your concerns. If there is any other concern that we could not address in the response, please feel free to let us know and we would be happy to provide further explanation.
>
> With sincere regards,
>
> Authors of Submission7387

---

### Meta-Review · Area_Chair_5oUC · 2026-01-05

**Summary:**

The paper introduces PPLLaVA, a novel prompt-guided pooling framework for video-based large language models to address the critical challenge of redundant visual tokens in both short and long videos. Initially, reviewers u53s and ddaz concern about architectural effectiveness, fairness of baselines, vague or generic situation and insufficient comparison with alternative compression methods, and reviewer eVAF argue that the author should conduct further experiments about model generality and token compression strategies, while reviewer Qq7T argue about the ablation about CLIP.

**Reviewer Concerns:**

The authors responded by clarifying that pooling weights are indirectly learnable via trainable text encoder, provided new ablations on InternVL3 and new results on Qwen2.5-VL, and reported FLOPs savings. These address most of the concorns. In addition, several issues remain partially addressed. Notably, they did not include direct comparisons with more token compression strategies, and no quantitative degradation analysis under vague prompts was added.

**Reviewer Scores:**

The four reviewers, especially with negative scores, will keep their scores, because some concerns are not well addressed.

---

### Decision · Program_Chairs · 2026-01-26

Accept (Poster)